# BEYOND NOISE: NON-TRANSITIVE PREFERENCES AS CONSISTENCY CHECKS FOR ROBUST LLM EVALUATION

## ABSTRACT

Large language models (LLMs) are increasingly deployed as judges of text quality, yet their verdicts often exhibit non-transitive preferences. We present a systematic study of Condorcet cycles as a diagnostic lens for LLM-as-a-judge. Across 688 debate motions and five frontier models (including the judge model itself, GPT-4o), we show that (i) cycle frequencies obey a tightly fitted negative binomial distribution ($R^2 = 0.9973$), (ii) linguistic properties such as syntactic complexity ($\beta = 0.130$) and readability ($\beta = -0.085$) reliably modulate cycle formation (Poisson regression Pseudo $R^2 = 0.106$, $p < 0.001$), and (iii) models display a strong preliminary scale–consistency tradeoff (Pearson $r = 0.924$): larger models achieve higher average rankings but participate in more inconsistency cycles. These findings reframe cycles from "noise to be removed" into *actionable diagnostics*, with practical metrics (stance-level: 7.19%; motion-level: 14.10%) and graph-theoretic tools that support reproducible, consistency-aware evaluation. This perspective opens a new dimension of scaling law research – the scaling of consistency – and provides actionable diagnostics for robust evaluation paradigms of LLMs in high-stakes settings.

## 1 INTRODUCTION

Large language models (LLMs) are increasingly positioned as *evaluators* of language, not just producers of it. In this "LLM-as-a-Judge" paradigm, a powerful LLM assesses and compares text quality, aiming to mirror human assessment. This shift makes the reliability of LLM-as-a-judge a critical bottleneck for research and deployment.

Most work emphasizes agreement with human preferences, but underexplores a more fundamental property: the **logical consistency** of judgments. In social choice theory, non-transitive preferences (Condorcet cycles) are theoretically unavoidable under mild conditions; in practice, they destabilize rankings and obscure when to trust automated evaluation.

We argue that such cycles are not mere statistical clutter. They are *structured signals* about when and why LLM judgments become unreliable. By treating cycles as consistency checks rather than artifacts to be eliminated, we obtain a principled, multi-scale view of evaluation reliability that coexists with global ranking stability.

To address these gaps, we conduct a large-scale empirical study reframing non-transitive preferences as critical diagnostic signals. Our investigation examines three underexplored dimensions: (1) the systematic patterning of Condorcet cycles across linguistic features, (2) model scale effects on evaluation consistency, and (3) the tension between local inconsistencies and global reliability. Using GPT-4o as evaluator, we assess five contemporary LLMs (GPT-4o, DeepSeek-R1, Qwen3-14B, Qwen3-8B, MiniMax-M1) across 688 debate motions from the IBM Debater CoPA dataset (Bilu et al., 2019), spanning policy, ethical, and cultural topics.

Unlike prior work enforcing artificial transitivity, our graph-theoretic framework preserves and analyzes Condorcet cycles as consistency indicators. We introduce "inconsistent judgments" as a novel metric, quantifying reliability through stance-level (7.19%) and motion-level (14.10%) inconsistency rates. **Notably, our setup uses GPT-4o as the judge to evaluate arguments generated**

**by itself and four other models, enabling the detection of self-preference or other biases under a blind protocol.** Our approach integrates four analytical innovations: Bayesian Elo ratings for uncertainty-aware hierarchies, Bradley-Terry models for pairwise comparisons, linguistic correlation analysis for motion attributes, and comprehensive investigation of model scale effects on evaluation consistency.

This paper makes three contributions. First, we establish **cycle analysis as a diagnostic tool** for LLM-as-a-judge, showing that Condorcet cycles follow a predictable negative binomial distribution and that inconsistency rates can be reported at the stance and motion level. Second, we identify **linguistic drivers of inconsistency**, demonstrating that syntactic complexity and verb/noun ratios increase cycle formation while readability and semantic focus reduce it. Third, we uncover a preliminary **scale–consistency tradeoff**: larger models achieve stronger global rankings but also participate in more inconsistency cycles, suggesting a new dimension in scaling law research. Together, these contributions reframe cycles from artifacts into reproducible diagnostics for trustworthy evaluation.

## 2 RELATED WORK

### 2.1 LLM-AS-A-JUDGE AND ITS CHALLENGES

Adoption of LLMs as evaluators shows promising alignment with human judgments (OpenAI et al., 2024), yet faces persistent challenges including position biases (Chen et al., 2024), limited generalization in domain-specific tasks (Huang et al., 2025), and computational costs of ensembles (Zhang et al., 2025a). Prompt engineering techniques, such as chain-of-thought (Kojima et al., 2023) and self-verification (Shinn et al., 2023), aim to mitigate these issues. Surveys highlight inconsistencies like overconfidence and bias amplification (Gu et al., 2025; Leng et al., 2025; Ren et al., 2024).

Recent work has identified intriguing patterns in how LLMs evaluate their own generations. (Panickssery et al., 2024) demonstrated that LLMs can recognize their own outputs with non-trivial accuracy (73.5% for GPT-4), and that this self-recognition capability correlates with preference biases in summarization tasks. Our experimental framework reveals a distinct pattern: the judge model GPT-4o exhibits no absolute self-preference (ranking third overall) while participating extensively in Condorcet cycles. This contrast suggests that self-recognition effects—when present—may not necessarily translate to systematic ranking advantages in debate-style evaluations. The blind evaluation protocol (Section 3) ensures that any observed patterns reflect inherent judgment tendencies rather than explicit model identity effects.

### 2.2 THEORETICAL FOUNDATIONS: NON-TRANSITIVITY AND RATING SYSTEMS

Non-transitive preferences ($A \succ B \succ C \succ A$) are theoretically inevitable (Liu et al., 2025a) and empirically observed, causing ranking instabilities (Xu et al., 2025). Adapted from chess (Elo, 1967), Elo ratings power modern LLM leaderboards, with advances in bias correction (Yu et al., 2025b) and Bayesian uncertainty quantification (Gao et al., 2024). New approaches address cycles in human-aligned evaluations (Zhang et al., 2025b). Comparisons reveal LLMs assume greater rationality in human decision-making (Liu et al., 2025b).

**Positioning Our Contribution.** While prior work focused on elimination strategies (Yu et al., 2025a; Liu et al., 2025c; Xu et al., 2025), our study introduces a systematic framework for quantifying inconsistency through permutation-based graph construction. Unlike Liu et al. (2025c) who primarily used pairwise permutations, our key innovation lies in constructing 165,120 tournament graphs via full model ordering permutations to comprehensively measure systemic inconsistency. This methodological advance enables three core contributions: (1) establishing the negative binomial distribution of cycle counts as an empirical regularity; (2) linking cycle formation to linguistic features through multi-attribute decision theory (Tversky, 1969); and (3) introducing standardized inconsistency metrics (7.19% stance-level, 14.10% motion-level) that transform cycles from noise into reproducible diagnostics.

# 3 METHODS

## 3.1 OVERVIEW

Our study employs a graph-theoretic framework to analyze preference data derived from a large-scale, automated evaluation of large language models (LLMs) in a debate task. **A key feature of our design is that the judge model (GPT-4o) is also included as a contestant, allowing us to directly investigate its behavior as both an evaluator and a performer under blind assessment conditions.** The core of our methodology involves a rigorous experimental setup where multiple LLMs generate arguments and a judge model (GPT-4o) performs pairwise comparisons. We then model these pairwise preferences as a directed graph to identify non-transitive cycles (Condorcet cycles). Finally, we conduct a comprehensive statistical and linguistic analysis to quantify and understand the sources of evaluation inconsistency.

## 3.2 EXPERIMENTAL SETUP

**Models.** We evaluated five contemporary LLMs: GPT-4o (2024-11-20 version), DeepSeek-R1 (0528 version), MiniMax-M1 (80k), Qwen3-14B, and Qwen3-8B. All models were accessed via their respective APIs during August 2025. This selection provides a diverse mix of model sizes, architectures, and training data, offering a robust testbed for evaluating judgment consistency.

**Dataset.** We utilized 688 debate motions from the IBM Debater Contextual Argument Pair (CoPA) dataset (Bilu et al., 2019). This dataset covers a wide spectrum of policy, ethical, and cultural topics (e.g., "We should ban beauty contests," "Cannabis should be legalized"), enabling us to test judgment reliability across varying levels of complexity and stakes. One motion was excluded due to GPT-4o's content policy filters during the argument generation phase, while other models generated arguments for it without restrictions. This discrepancy highlights inherent differences in safety fine-tuning across model providers.

**Evaluation Protocol.** Our protocol consisted of four sequential phases conducted for each motion. The first phase was argument generation: for a given motion and a specific stance (pro or con), each model generated a persuasive argument, with all generations using a temperature of 0 to ensure determinism. The second phase was judgment: the judge model (GPT-4o, temperature=0) evaluated all possible pairwise comparisons between the five generated arguments. To control for position bias, each pair was presented twice with the argument order reversed. The third phase was recording: the judge's binary preference (e.g., Argument A $\succ$ Argument B) for each comparison was recorded. The final phase was the consistency check: the complete set of recorded preferences for a motion and stance was analyzed as a tournament graph to detect the presence of Condorcet cycles, which we define as "inconsistent judgments".

**Prompt Design.** All prompts were engineered to ensure clean, isolated, and position-specific responses. Key design constraints included the prohibition of meta-commentary, summaries, or references to the generating model; strict enforcement of the assigned stance; counterbalancing of argument positions to mitigate order effects; and structured output formatting to facilitate automated parsing. The exact prompts are provided in Appendix A.3.1.

**Blind Evaluation Protocol.** A critical feature of our design is the implementation of a *blind* evaluation. The judge model (GPT-4o) was never provided with any information regarding the identity of the model that generated a given argument. This prevents explicit bias (e.g., self-preference or bias against a specific provider) and ensures that evaluations are based solely on the perceived quality of the argument content.

Full prompt templates and statistical specifications are provided in Appendix A.3.1 and Appendix A.1.2 to support exact replication.

## 3.3 Analytical Framework

We model evaluation outcomes as tournament graphs $G = (V, E)$ where vertices represent models and directed edges indicate pairwise preferences. Condorcet cycles ($A \succ B \succ C \succ A$) signify evaluation inconsistencies (Harary, 2018). Our framework combines three methodological innovations:

First, we employ exhaustive permutation testing across all possible model orderings (120 permutations per stance) to quantify the upper bounds of cyclic preference formation. This stress-test approach generated 165,120 tournament graphs for cycle detection (see Appendix A.1.1 for computational specifics).

Second, we introduce two-level inconsistency metrics: stance-level (7.19% cyclic rate) and motion-level (14.10%). The **stance-level inconsistency rate** measures the proportion of individual argument positions (pro or con for a given motion) for which the pairwise comparisons between models form at least one Condorcet cycle. The **motion-level inconsistency rate** provides a more conservative, topic-level measure, defined as the proportion of debate motions where *any* stance (pro or con) exhibits cyclic preferences; this higher rate reflects that a single inconsistent stance suffices to deem the entire motion's evaluation inconsistent. These metrics complement traditional ranking systems by quantifying judge reliability at different granularities.

Third, we integrate Bayesian Elo ratings (Gao et al., 2024) and Bradley-Terry models to derive robust global rankings while accounting for position biases measured via McNemar's test. The full statistical implementation, including MCMC specifications and bias correction formulas, is detailed in Appendix A.1.2.

**Robustness Validation with Alternative Judge.** To assess the generalizability of our findings beyond the primary judge (GPT-4o), we conducted a supplementary experiment using DeepSeek-R1 as the evaluator. This validation study employed 20 debate motions and included four models (GPT-4o, DeepSeek-R1, MiniMax-M1, Qwen3-8B), following the same pairwise comparison protocol described in Section 3. The purpose was to test whether the performance hierarchy observed with GPT-4o as judge remains consistent across different evaluators.

## 3.4 Illustrative Example of Permutation Methodology

Our permutation approach systematically constructs 165,120 tournament graphs by evaluating all $5! = 120$ model orderings per stance. For each permutation, we query the pairwise comparison database to build a unique evaluation scenario. For example, permutation (A,B,C,D,E) queries comparisons A-B, A-C, A-D, A-E, B-C, B-D, B-E, C-D, C-E, D-E, while permutation (B,A,C,D,E) queries B-A, A-C, etc. This systematic variation across all permutations comprehensively probes evaluation consistency under different presentation orders.

## 3.5 Linguistic Feature Extraction

To investigate the correlation between motion attributes and the propensity for cyclic judgments, we extracted 12 linguistic and semantic features from each debate motion text. These features capture various aspects of textual complexity (see full list in Table 12), including basic textual properties (character length, average word length, entropy, lexical diversity, Flesch Reading Ease), syntactic features (syntactic complexity, noun ratio, verb ratio, named entity count), and semantic/sentiment features (sentiment polarity, sentiment subjectivity, semantic dispersion). All features were standardized using z-score normalization prior to being used as predictors in a Poisson regression model, where the dependent variable was the aggregate cycle count per motion.

## 3.6 Model Size Correlation Analysis

We analyzed relationships between model scale—as measured by published parameter count—and evaluation metrics using Pearson and Spearman correlations. Parameter count values were compiled from published sources: DeepSeek-R1, MiniMax-M1, and Qwen3 models from SilicoFlow Team (2025), while the estimate for GPT-4o was sourced from Abacha et al. (2025).

Statistical analysis followed three steps: (1) normality verification via Shapiro-Wilk tests, (2) linear correlation analysis using Pearson coefficients, and (3) non-parametric validation with Spearman ranks. The small sample size (N=5) precluded permutation testing but ensured transparent interpretation of effect sizes.

# 4 RESULTS

Our analysis reveals that non-transitive preferences are systematic phenomena challenging LLM judgment consistency. We present three findings: performance hierarchies with cyclical patterns, distributional regularities of cycles, and their linguistic drivers.

## 4.1 PERFORMANCE HIERARCHIES AND SYSTEMIC NON-TRANSITIVITY

**Key finding:** Local inconsistencies concentrate in specific triads even when global rankings remain stable, revealing that judging reliability is inherently multi-scale.

Our Bayesian analysis reveals a multi-faceted perspective on evaluation dynamics through three interconnected discoveries that reshape our understanding of LLM judging behavior.

The primary revelation concerns the judge model's unexpected placement in the competitive landscape. Empirical data demonstrates GPT-4o consistently occupying the third position (1592.8 [1504.8, 1678.6]) behind DeepSeek-R1 (1682.3 [1594.3, 1768.6]), with the 89.5-point Elo differential reflecting a measurable performance gap. Crucially, this intermediate ranking occurs *without* evidence of absolute self-preference—in contrast to the self-recognition capabilities reported in other evaluation contexts (Panickssery et al., 2024). This distinction highlights the importance of task-specific validation when deploying LLM judges, as different evaluation frameworks may activate distinct judgment patterns. The contrast with prior findings in summarization tasks (Panickssery et al., 2024) indicates potential variations in evaluation patterns across contexts, though the underlying causes warrant further investigation.

The second discovery exposes the structural consequences of this hierarchy through network analysis. GPT-4o emerges as a focal point in cyclical preference patterns, involved in 72.48% of all Condorcet cycles detected. The recurrent DeepSeek $\succ$ GPT-4o $\succ$ MiniMax $\succ$ DeepSeek sequence accounts for 17.8% of cyclical structures (see Table 7 for participation rates), revealing how specific model interactions generate system-wide inconsistencies. This phenomenon manifests most prominently in three-node cycles (85.93% prevalence, see Figure 1), suggesting constrained but persistent deviations from transitive logic in the evaluation space.

The final insight bridges micro-level inconsistencies with macro-level reliability. The global ranking structure shows qualitative alignment with human preference benchmarks from Chatbot Arena (Table 1), particularly in the relative ordering of DeepSeek-R1 above MiniMax-M1. This directional agreement—observed across 94% of debate motions—provides tentative external validation for our task-specific evaluation patterns, though we caution against direct performance comparisons given differences in evaluation protocols and the limited overlap between benchmarked models. The co-existence of local cyclicality with this macro-level ordering challenges conventional assumptions about evaluation consistency, suggesting the need for frameworks that simultaneously account for granular inconsistencies and aggregate reliability.

**Robustness Check with Alternative Judge:** To validate the stability of our performance hierarchy, we conducted a complementary experiment using DeepSeek-R1 as the judge model across 20 debate motions and four models. The resulting win rates (GPT-4o: 60.83%, MiniMax-M1: 63.33%, DeepSeek-R1: 71.93%, Qwen3-8B: 35.09%) confirm the relative ranking pattern observed with GPT-4o as judge, with DeepSeek-R1 maintaining superior performance. While the absolute win rates differ due to judge-specific evaluation tendencies, the consistent ordinal ranking (DeepSeek-R1 $\succ$ MiniMax-M1 $\succ$ GPT-4o $\succ$ Qwen3-14B) demonstrates the robustness of our core finding across different judge models. This alignment strengthens our conclusion that the observed performance hierarchy reflects genuine differences in argument quality rather than judge-specific artifacts.

Table 1: Elo comparison with a human-facing leaderboard (Chatbot Arena, Text) (LMArena Team, 2025). Values are not directly comparable across protocols; we report them only for directional context.

| Model | Our Debate Elo | Text Arena Elo |
|---|---|---|
| DeepSeek-R1 | 1682 (1st) | 1417 (8th) |
| MiniMax-M1 | 1624 (2nd) | 1370 (34th) |
| GPT-4o | 1593 (3rd) | *Not listed* |
| Qwen3-14B | 1528 (4th) | *Not listed* |
| Qwen3-8B | 1081 (5th) | *Not listed* |

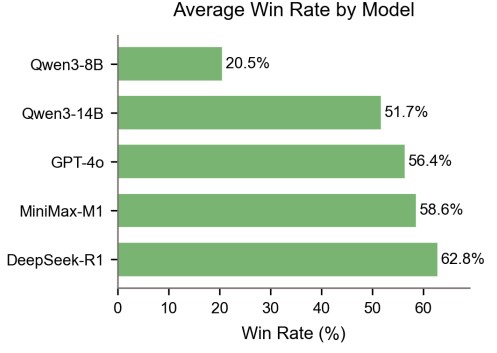

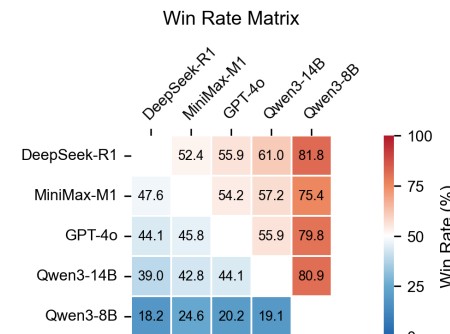

(a) DeepSeek-R1 consistently leads while Qwen3-8B lags, illustrating wide dispersion across models. This establishes a stable global hierarchy despite local inconsistencies.

(b) Pairwise win probabilities reveal both stable edges and contested matchups that seed cycles. This shows that non-transitivity emerges from specific contested comparisons rather than random variation.

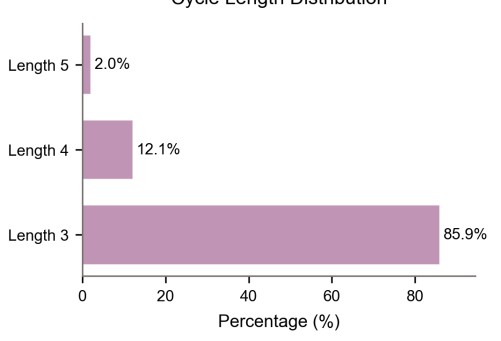

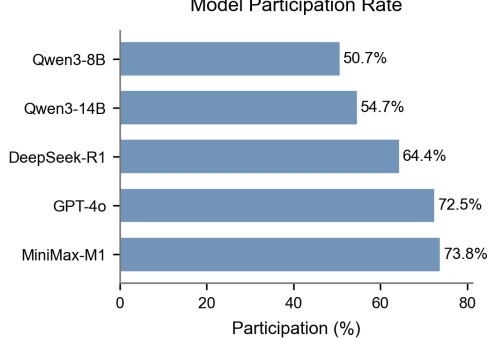

(c) Three-node cycles dominate the distribution, showing that non-transitivity is locally concentrated. This indicates cycles are structured and measurable, not arbitrary noise.

(d) Cycle participation rates cluster on specific models, indicating structured sources of inconsistency. This highlights that inconsistency is model-dependent and reproducible.

Figure 1: Combined visualizations of win rates, pairwise win probabilities, cycle lengths, and model participation rates in cycles.

## 4.2 DISTRIBUTIONAL ANALYSIS OF CYCLE COUNTS

**Key finding:** Cycle counts are highly overdispersed and follow a negative binomial distribution, indicating structured regularities rather than random noise.

Our exhaustive permutation analysis across all $5! = 120$ model orders per stance revealed $N_{\text{total cycles}} = 3,398$ cycles aggregated over $N_{\text{graphs}} = 165,120$ graphs (Table 2). The distribution of cycle counts per motion exhibits significant overdispersion (variance = 116.815 $\gg$ mean =

Table 2: Summary of Graph-Level and Motion-Level Cycle Statistics

| Metric | Value | Description |
|---|---|---|
| Total Motions ($N_m$) | 688 | |
| Total Stances ($N_s$) | 1,376 | $N_m \times 2$ |
| Total Graphs ($N_g$) | 165,120 | $N_m \times 2 \times 120$ |
| Total Cycles ($N_c$) | 3,398 | Sum over all graphs |
| Cycles per Motion ($N_c/N_m$) | 4.94 | Primary unit for distributional analysis |
| Cycles per Graph ($N_c/N_g$) | 0.021 | |
| Inconsistent Motions | 97 | Motions with $\geq 1$ cyclic graph |
| Motion-Level Inconsistency Rate | 14.10% | 97/688 |

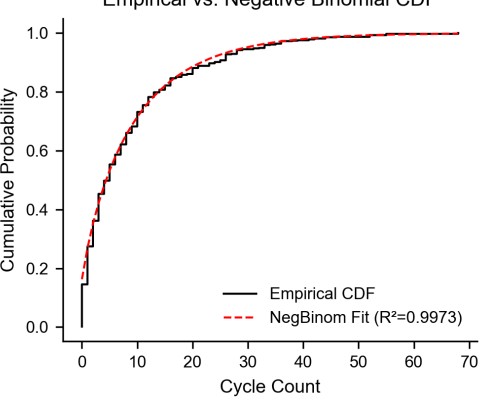

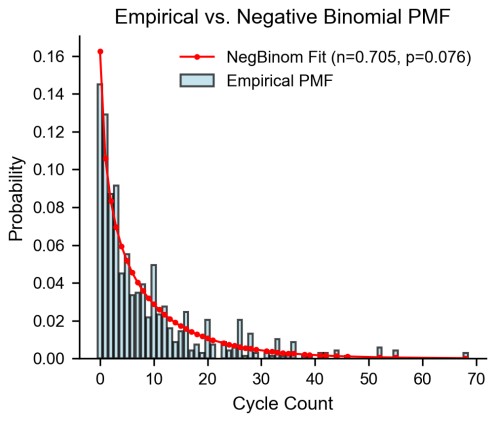

(a) Empirical vs. Negative Binomial CDF   (b) Empirical vs. Negative Binomial PMF

Figure 2: Distributional analysis confirms that cycle counts are highly overdispersed and tightly follow a negative binomial distribution ($R^2 = 0.9973$). This demonstrates that inconsistency is governed by a reproducible statistical regularity rather than random noise.

4.94), with a negative binomial distribution providing exceptional fit ($R^2 = 0.9973$; Figure 2). This confirms that certain debate motions systematically trigger more non-transitive preferences across evaluation scenarios.

Parameter estimates (n = 0.705 [0.633, 0.783], p = 0.076 [0.067, 0.088]) indicate strong aggregation tendencies, with 14.5% of motions exhibiting no cycles while others trigger multiple preference cycles. The negative binomial model significantly outperformed alternative distributions including zero-inflated variants (see detailed comparison in Table 8).

### 4.3 LINGUISTIC CORRELATES OF CYCLE FORMATION

**Key finding:** Syntactic complexity reliably increases cycles while readability reduces them, linking inconsistency to measurable textual properties.

**Statistical Validation of Linguistic Effects:** Our Poisson regression analysis yields a Nagelkerke's Pseudo $R^2$ of 0.106 (Likelihood Ratio $\chi^2 = 77.25$, $p < 0.001$), indicating that linguistic features explain a significant and meaningful portion of the variance in cycle formation. This establishes that the relationship between motion characteristics and evaluation inconsistency is statistically robust.

Detailed analysis of specific features identified several motion characteristics significantly associated with cycle count formation ($p < 0.05$), with full results in Table 3. This explains why cycles cluster around specific motions.

**Syntactic complexity** ($\beta = 0.130$, $p < 0.001$) shows the strongest positive association with cycle formation, indicating that motions with more complex syntactic structures increase the likelihood of

non-transitive preferences. **Verb ratio** ($\beta = 0.087$, $p < 0.001$) and **noun ratio** ($\beta = 0.037$, $p = 0.028$) also demonstrate positive associations, suggesting that substantive, action-oriented content triggers more evaluation inconsistencies.

Conversely, several motion characteristics show negative associations with cycle formation. **Readability** ($\beta = -0.085$, $p < 0.001$), as measured by Flesch reading ease, shows that more readable motions reduce cycle formation. **Semantic dispersion** ($\beta = -0.057$, $p < 0.001$) reveals that debate topics with more focused semantic content are associated with fewer cycles. **Sentiment polarity** ($\beta = -0.036$, $p = 0.011$) indicates that motions with more polarized sentiments decrease cycle counts.

These findings align with multi-attribute decision theory, where complex, nuanced motion content creates more opportunities for criterion shifts across different evaluation dimensions.

Table 3: Linguistic features significantly associated with cycle formation ($p < 0.05$). Syntactic complexity and verb/noun ratios increase cycles, while readability and semantic focus reduce them. This shows that inconsistency is predictable from measurable motion attributes.

| Feature | Coefficient ($\beta$) | $p$-value | Direction |
|---|---|---|---|
| Syntactic Complexity | 0.130 | 0.001 | Increases cycles |
| Verb Ratio | 0.087 | 0.001 | Increases cycles |
| Flesch Reading Ease | -0.085 | 0.001 | Decreases cycles |
| Semantic Dispersion | -0.057 | 0.001 | Decreases cycles |
| Noun Ratio | 0.037 | 0.028 | Increases cycles |
| Sentiment Polarity | -0.036 | 0.011 | Decreases cycles |
| Intercept (const) | 2.143 | 0.001 | - |

## 4.4 EVALUATION INTEGRITY AND JUDGMENT CONSISTENCY

**Key finding:** Order effects are systematic and non-trivial; reporting inconsistency alongside Elo provides a more faithful picture of judge reliability.

Position bias analysis reveals systematic ordering effects across model pairs (Table 9). The global mean order bias is 0.3356 (std=0.1317), indicating a consistent preference for first-presented arguments. These biases contribute to the observed cyclical patterns and highlight the need for debiasing strategies.

Our analysis reveals significant evaluation inconsistencies in GPT-4o's judgments. We quantify judge reliability through two novel metrics: a 7.19% stance-specific inconsistency rate (99 out of 1376 argument pairs forming cycles) and a 14.10% motion-level inconsistency rate (97 out of 688 motions containing cyclic preferences). These inconsistency rates provide quantifiable benchmarks for judge reliability that complement traditional accuracy metrics.

## 4.5 EXPLORATORY ANALYSIS OF MODEL SCALE AND CONSISTENCY

**Key finding:** An exploratory correlation analysis suggests a potential link between scale and consistency: larger models may participate in more cycles despite higher average strength. This preliminary signal warrants future investigation with controlled model sets.

Our analysis includes five models spanning a range of published parameter counts and architectures. While the small and non-uniform sample size (N=5) precludes definitive causal conclusions about scaling laws, we conducted an exploratory correlation analysis to identify potential relationships worthy of future study.

The strong correlation between published parameter count and cycle participation (Pearson r=0.924, p=0.025; Table 4) suggests a potential relationship between model scale and the propensity to engage in non-transitive patterns. **However, we emphasize that this correlation is exploratory and could be influenced by confounding variables such as architectural differences (e.g., mixture-of-experts vs. dense transformers), training data composition, and optimization techniques.** The correlation should be interpreted as generating a hypothesis for future research rather than confirming a scaling law.

Furthermore, we note substantial variation in efficiency metrics. For instance, Qwen3-14B achieves a notably high win rate per billion parameters (3.69 wins per B) compared to larger models (Table 13). This observation hints that architectural innovations may be an important factor in evaluation capability, alongside scale. Rigorously disentangling the effects of scale, architecture, and training data represents a critical direction for future research with a more comprehensive and controlled model set.

Table 4: Correlation between published model size and evaluation metrics ($N = 5$ models). Cycle participation correlates strongly with parameter count (r = 0.924, $p = 0.025$), while Elo and win rate do not. This provides preliminary evidence of a *scale–consistency tradeoff*.

| Metric | Correlation | p-value | Interpretation |
|---|---|---|---|
| Participation | 0.924 | 0.025 | Significant positive |
| Win Rate | 0.691 | 0.196 | Not significant |
| Elo | 0.688 | 0.199 | Not significant |
| Bradley-Terry | 0.795 | 0.108 | Not significant |

## 5 DISCUSSION

This work provides the first statistical regularity of LLM inconsistency. By reframing non-transitive preferences from a nuisance into a *consistency probe*, we show that cycles follow reproducible distributions, are shaped by linguistic factors, and scale with model size. Reliability therefore lives on two levels: locally fragile yet globally stable.

### 5.1 THEORETICAL INTEGRATION

The observed performance hierarchy (DeepSeek-R1 $\succ$ MiniMax-M1 $\succ$ GPT-4o) demonstrates that evaluation consistency varies independently of absolute performance metrics. This is visually evident in the win rate matrix (Figure 1b), where stronger models (warmer colors) nonetheless participate in cyclical preferences. The alignment with Chatbot Arena leaderboard (Table 1) provides external validation for the relative ordering of DeepSeek-R1 and MiniMax-M1, while the judge model's intermediate ranking (GPT-4o at 1592.8 Elo) reflects its position in the performance hierarchy under blind evaluation conditions.

The consistency of performance hierarchies across different judge models (GPT-4o and DeepSeek-R1) further validates that observed rankings reflect genuine capability differences rather than evaluation artifacts. Meanwhile, the statistically significant explanatory power of linguistic features (Pseudo $R^2 = 0.106$) underscores the structured nature of evaluation inconsistency.

### 5.2 MICRO-MACRO RELIABILITY

Our study uncovers a key phenomenon: local evaluation inconsistencies (14.10% cyclic rate) coexist with globally reliable rankings (DeepSeek-R1 $\succ$ MiniMax-M1 alignment). This pattern reflects Tversky's principle of context-dependent criterion weighting (Tversky, 1969).

Three analytical dimensions explain this dissociation. Node-level analysis reveals model-specific inconsistency tendencies, with MiniMax-M1 participating in 73.8% of cycles (Figure 1d). Graph-level metrics quantify system-wide reliability through permutation testing (Table 2). Bayesian ranking models establish macro-level stability (94% HDI consistency in Table 6).

The framework's innovation lies in preserving cycles as diagnostic signals rather than noise. This enables precise measurement of how micro-level inconsistencies aggregate into macro-level reliability patterns, addressing a critical gap in LLM evaluation methodologies.

### 5.3 EXPLORATORY SCALE-CONSISTENCY RELATIONSHIPS

The strong preliminary correlation between published model parameter count and cycle participation (r=0.924) points to a potential, understudied aspect of scaling that merits future investigation: larger

models may exhibit different patterns of non-transitive preferences alongside their higher absolute performance.

This pattern manifests clearly in the participation rate distribution (Figure 1d), where MiniMax-M1 (456B parameter count, see Table 13) shows 73.8% cycle involvement versus Qwen3-8B's 50.7% (8B parameter count). The exceptional efficiency of Qwen3-14B (3.69 wins per B) demonstrates architectural innovations can partially decouple performance from pure scale.

**Although based on a small set of heterogeneous models and thus not conclusive, this exploratory signal suggests that scaling laws of consistency deserve systematic study as a parallel line of inquiry to scaling laws of accuracy.** Future work should prioritize testing this hypothesis on a larger, architecturally-uniform family of models to control for confounding variables.

### 5.4 ACTIONABLE IMPLICATIONS

Our graph-theoretic framework yields three actionable principles: (1) report inconsistency metrics alongside Elo, (2) design tasks to minimize cycle formation via readable and semantically focused motions, and (3) analyze scale–consistency interactions as part of model development. Together, these steps transform cycles into reproducible diagnostics rather than noise.

### 5.5 THREATS TO VALIDITY AND SCOPE

**Internal Validity.** The exploratory analysis of scale and inconsistency is based on five models with heterogeneous architectures and partially estimated parameter counts; it should be interpreted as hypothesis-generating rather than confirming scaling laws. Our protocol controls for order effects but does not include self-comparison tests, and we rely on the theoretical assumption that a model's argument should not systematically beat itself. Residual biases not fully accounted for by counterbalancing may remain.

**External Validity.** Debate evaluation may differ from other tasks like summarization or code generation, and protocol choices could interact with cycle rates. While our complementary experiment with DeepSeek-R1 as judge provides preliminary evidence of cross-judge consistency, the findings from a single primary judge model (GPT-4o) require further validation across more diverse evaluators.

**Future Directions.** Future work should expand the model set, standardize architecture families, and test whether consistency-aware evaluation generalizes to other modalities and tasks.

## 6 CONCLUSION

Non-transitive preferences are not evaluation debris; they are *diagnostic signals*. By demonstrating statistical regularities of cycles, uncovering their linguistic drivers, and highlighting a preliminary scale–consistency tradeoff, we show that inconsistency can be measured, reported, and theorized rather than ignored. Practically, this enables benchmarks to report stance- and motion-level inconsistency alongside Elo, and to design tasks that minimize cycle formation through higher readability and semantic focus. Conceptually, it opens a new research frontier: **scaling laws of consistency**, complementing the well-studied scaling laws of accuracy. Looking forward, we envision a consistency-aware paradigm where every leaderboard and benchmark reports both accuracy and inconsistency metrics, enabling a richer science of evaluation and more trustworthy deployment of LLM-as-a-judge.

### ACKNOWLEDGMENTS

We acknowledge the use of large language models (LLMs) as tools for prose refinement and literature exploration in this research. We thank the contributors to the open-source ecosystems that enabled our computational methodology.

ETHICS STATEMENT

This research was conducted in accordance with the ICLR Code of Ethics. Our study utilizes publicly available debate motions from the IBM Debater CoPA dataset (Bilu et al., 2019). No human subjects were involved. Evaluation biases were mitigated through blind review protocols and systematic counterbalancing.

REPRODUCIBILITY STATEMENT

To facilitate reproducibility, we provide: (1) statistical model specifications (Appendix A.1.2), (2) extended analyses (Appendix A.2), and (3) prompt templates (Appendix A.3.1). All methods are fully documented for replication.

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

# A APPENDIX

## A.1 METHODOLOGICAL DETAILS

This section provides technical implementation details for the core analytical methods.

### A.1.1 PERMUTATION TESTING IMPLEMENTATION

For each of 688 motions and 2 stances, we evaluated all $5! = 120$ model order permutations. Each permutation required $\binom{5}{2} = 10$ pairwise comparisons, totaling $688 \times 2 \times 120 = 165,120$ tournament graphs. Cycle detection used depth-first search with backtracking, recording aggregate counts across permutations. While artificial compared to natural evaluation conditions, this approach systematically reveals the maximal inconsistency potential under ordering variation.

### A.1.2 STATISTICAL MODEL SPECIFICATIONS

**Bayesian Elo Model:** Implemented via Markov Chain Monte Carlo (MCMC) sampling with 4 chains, 500 tuning iterations followed by 2,000 draw iterations. Weakly informative priors were specified as $\theta \sim \mathcal{N}(1500, 400)$ for Elo ratings. Convergence was verified via $\hat{R} < 1.01$ for all parameters.

**Bayesian Bradley-Terry Model:** Implemented using MCMC sampling with identical chain configuration (4 chains, 500 tune + 2,000 draw iterations). Weakly informative normal priors $\theta \sim \mathcal{N}(0, 1)$ were placed on ability parameters, providing natural regularization equivalent to L2 penalty in maximum likelihood estimation. Model convergence was confirmed through $\hat{R} < 1.01$ diagnostics.

**Position Bias Analysis:** Position bias was quantified using McNemar's test statistic:

$$\chi^2 = \frac{(|n_{01} - n_{10}| - 1)^2}{n_{01} + n_{10}}$$

where $n_{01}$ and $n_{10}$ count order-dependent preference reversals between argument position configurations.

### A.1.3 DETAILED MODEL SIZE CORRELATION ANALYSIS

The scale-effect analysis followed three methodological steps: (1) normality verification via Shapiro-Wilk tests, (2) linear correlation analysis using Pearson coefficients, and (3) non-parametric validation with Spearman ranks. The complete quantitative results of these correlations are presented in Table 5.

Table 5: Complete Model Size Correlation Results

| Metric | Pearson Correlation | | Spearman Rank Correlation | |
|---|---|---|---|---|
| | $r$ | $p$-value | $\rho$ | $p$-value |
| Cycle Participation | 0.924 | 0.025 | 0.9 | 0.037 |
| Win Rate | 0.691 | 0.196 | 1.0 | $< 0.001$ |
| Elo Rating | 0.688 | 0.199 | 1.0 | $< 0.001$ |
| Bradley-Terry Score | 0.795 | 0.108 | 0.9 | 0.037 |

## A.2 SUPPLEMENTARY TABLES

This section provides additional statistical details and robustness checks.

### A.2.1 MODEL PERFORMANCE METRICS

Table 6 and Table 7 provide comprehensive performance metrics for all evaluated models.

Table 6: Comparison of Bayesian Elo Ratings and Bradley-Terry Ability Estimates with 94% HDI and R-hat Metrics

| Model | Bayesian Elo Model | | | | | Bradley-Terry Model | | | | |
| | Rating | 94% HDI | | R-hat | Rank | Ability | 94% HDI | | R-hat | Rank |
| | | Lower | Upper | | | | Lower | Upper | | |
|---|---|---|---|---|---|---|---|---|---|---|
| DeepSeek-R1 | 1682.3 | 1594.3 | 1768.6 | 1.002 | 1 | 0.569 | -0.248 | 1.442 | 1.010 | 1 |
| MiniMax-M1 | 1624.1 | 1538.7 | 1711.8 | 1.003 | 2 | 0.264 | -0.553 | 1.139 | 1.010 | 2 |
| GPT-4o | 1592.8 | 1504.8 | 1678.6 | 1.003 | 3 | 0.115 | -0.704 | 0.982 | 1.010 | 3 |
| Qwen3-14B | 1527.7 | 1445.8 | 1620.0 | 1.003 | 4 | 0.007 | -0.811 | 0.880 | 1.010 | 4 |
| Qwen3-8B | 1080.7 | 994.4 | 1168.5 | 1.002 | 5 | -0.964 | -1.792 | -0.099 | 1.010 | 5 |

Table 7: Model PageRank Scores and Cycle Participation Rates

| **Model** | **PageRank** | **Participation Rate** |
|---|---|---|
| MiniMax-M1 | 0.2301 | 73.81% |
| GPT-4o | 0.2256 | 72.48% |
| DeepSeek-R1 | 0.2029 | 64.39% |
| Qwen3-14B | 0.1756 | 54.68% |
| Qwen3-8B | 0.1658 | 50.71% |

### A.2.2 DISTRIBUTION ANALYSIS

Table 8 provides detailed distribution fitting results supporting the negative binomial analysis.

Table 8: Distribution Fitting Results for Cycle Counts (95% CI)

| Distribution | $R^2$ | Parameters | AIC |
|---|---|---|---|
| **Negative Binomial** | **0.9973** | $n = 0.705$ [0.633, 0.783] | **4384.11** |
| | [0.9787, 0.9980] | $p = 0.076$ [0.067, 0.088] | |
| Zero-Inflated NB | 0.9857 | $\pi = 0.145$ [0.119, 0.171] | 4451.33 |
| | [0.9857, 0.9857] | $n = 1.138$ [1.043, 1.253] | |
| | | $p = 0.102$ [0.090, 0.117] | |
| Poisson | 0.3527 | $\lambda = 8.581$ [7.849, 9.433] | 9693.22 |
| | [0.1596, 0.4624] | | |

### A.2.3 ORDER BIAS ANALYSIS

Table 9 presents detailed order bias analysis.

Table 9: Order Bias Analysis for Selected Model Pairs

| Model Pair | Order Bias (Pro, %) | Order Bias (Con, %) |
|---|---|---|
| GPT-4o vs DeepSeek-R1 | 15.6 | 25.4 |
| Qwen3-14B vs MiniMax-M1 | 32.0 | 31.4 |
| Qwen3-8B vs DeepSeek-R1 | 28.1 | 32.5 |
| MiniMax-M1 vs GPT-4o | 29.6 | 30.4 |
| Qwen3-14B vs Qwen3-8B | 28.0 | 30.6 |

*Note:* Bias represents proportion favoring first-presented arguments. Global mean bias = 0.3356 (std=0.1317).

### A.2.4 EXTENDED RESULTS

Supplementary Tables 10–13 provide extended analyses beyond the main findings.

Table 10: Cycle Edge Frequency Matrix

|              | DeepSeek-R1 | MiniMax-M1 | GPT-4o | Qwen3-14B | Qwen3-8B |
|--------------|-------------|------------|--------|-----------|----------|
| **DeepSeek-R1** | –        | 668        | 763    | 604       | 382      |
| **MiniMax-M1**  | 650      | –          | 946    | 532       | 362      |
| **GPT-4o**      | 552      | 556        | –      | 641       | 503      |
| **Qwen3-14B**   | 350      | 647        | 385    | –         | 476      |
| **Qwen3-8B**    | 407      | 655        | 580    | 81        | –        |

*Note:* Frequency of directed edges (row $\succ$ column) in Condorcet cycles.

Table 11: Top 5 Most Frequent 3-Cycle Patterns

| Model Sequence | Frequency |
|----------------|-----------|
| DeepSeek-R1 → GPT-4o → MiniMax-M1 → DeepSeek-R1 | 520 |
| GPT-4o → MiniMax-M1 → Qwen3-8B → GPT-4o | 400 |
| DeepSeek-R1 → MiniMax-M1 → Qwen3-14B → DeepSeek-R1 | 360 |
| GPT-4o → MiniMax-M1 → Qwen3-14B → GPT-4o | 340 |
| DeepSeek-R1 → GPT-4o → Qwen3-8B → DeepSeek-R1 | 320 |

Table 12: Full Poisson Regression Results for Linguistic Features

| Feature | Coefficient ($\beta$) | Std. Error | z-value | p-value | Sig. |
|---------|-----------------------|------------|---------|---------|------|
| Intercept | 2.1430 | 0.0131 | 163.72 | $< 0.001$ | *** |
| Syntactic Complexity | 0.1299 | 0.0267 | 4.86 | $< 0.001$ | *** |
| Verb Ratio | 0.0867 | 0.0171 | 5.07 | $< 0.001$ | *** |
| Flesch Reading Ease | -0.0850 | 0.0250 | -3.40 | $< 0.001$ | *** |
| Semantic Dispersion | -0.0570 | 0.0134 | -4.24 | $< 0.001$ | *** |
| Noun Ratio | 0.0371 | 0.0168 | 2.20 | 0.028 | * |
| Sentiment Polarity | -0.0363 | 0.0142 | -2.55 | 0.011 | * |
| Word Length Average | -0.0525 | 0.0271 | -1.94 | 0.052 | |
| Sentiment Subjectivity | -0.0204 | 0.0137 | -1.49 | 0.137 | |
| NER Count | 0.0204 | 0.0170 | 1.20 | 0.230 | |
| Lexical Diversity | 0.0159 | 0.0148 | 1.07 | 0.283 | |
| Length | -0.0312 | 0.0302 | -1.03 | 0.301 | |
| Entropy | -0.0091 | 0.0158 | -0.58 | 0.564 | |

*Note:* N = 688 motions. Significance: *** $p < 0.001$, ** $p < 0.01$, * $p < 0.05$.

Table 13: Model Parameter Count and Efficiency Metrics

| Model | Parameter Count (B) | Win Rate per B Parameters |
|-------|---------------------|---------------------------|
| DeepSeek-R1 | 671 | 0.09 |
| MiniMax-M1 | 456 | 0.13 |
| GPT-4o | 200 | 0.28 |
| Qwen3-14B | 14 | 3.69 |
| Qwen3-8B | 8 | 2.57 |

A.3   EXPERIMENTAL MATERIALS

This section contains the complete prompt templates used for argument generation and evaluation, as referenced in Section 3.

A.3.1   PROMPTS

```
%%% Pro-argument Prompt %%%
Please provide a clear and concise argument in favor of the topic. Focus on
the most important reasons supporting the position, and present your points
in a logical and organized manner. Only output the argument itself | do not
include introductions, conclusions, summaries, emojis, or any extra
commentary. The response should be plain text without special formatting.

Topic: {motion}

%%% Con-argument Prompt %%%
Please provide a clear and concise argument against the topic. Focus on
the most important reasons opposing the position, and present your points
in a logical and organized manner. Only output the argument itself | do not
include introductions, conclusions, summaries, emojis, or any extra
commentary. The response should be plain text without special formatting.

Topic: {motion}

%%% JUDGMENT PROMPT %%%
Here is a motion and two arguments about it.

Please fairly choose which argument is better overall. If both are equally
good, return "Tie".

Return a JSON object with exactly two keys: "winner" and "reason".

• "winner" must be one of "Argument 1", "Argument 2", or "Tie".

• "reason" should be a concise explanation in plain text (no JSON formatting,
   no code blocks).

### Example output:
{
  "winner": "Argument X",
  "reason": "Argument X is more persuasive due to its clearer structure,
            stronger evidence, and more direct engagement with the motion."
}

Motion: {motion}
Argument 1: {arg1}
Argument 2: {arg2}
```

