# OpenReview forum: "Beyond Noise: Non-Transitive Preferences as Consistency Checks for Robust LLM Evaluation"
_ICLR.cc/2026/Conference — Submitted to ICLR 2026_

### Official Review · Reviewer_h7Gd · 2025-10-27

**Soundness:** 2
**Presentation:** 3
**Contribution:** 3
**Rating:** 6
**Confidence:** 3

**Summary:**

This paper presents a systematic investigation of non-transitive preferences (Condorcet cycles) in LLM-as-a-judge evaluation systems. Rather than treating these cycles as noise to be eliminated, the authors reframe them as diagnostic signals for evaluation reliability. Through analysis of 688 debate motions across five contemporary LLMs using GPT-4o as judge, they demonstrate that cycle frequencies follow a negative binomial distribution ($R^2 = 0.9973$), correlate with linguistic features such as syntactic complexity and readability, and exhibit a preliminary scale-consistency tradeoff where larger models participate in more cycles despite higher performance. The work introduces stance-level (7.19%) and motion-level (14.10%) inconsistency metrics as complementary measures to traditional ranking systems.

**Strengths:**

The paper makes a conceptually interesting contribution by reframing non-transitive preferences from artifacts to be removed into structured diagnostic signals. This perspective shift is valuable for the community as it provides a more nuanced understanding of LLM judge reliability. The statistical rigor is commendable, particularly the demonstration that cycle counts follow a negative binomial distribution with exceptional fit. This finding suggests that inconsistency is not random but follows reproducible statistical regularities, which is an important theoretical insight.

The experimental design is well thought out, especially the blind evaluation protocol where GPT-4o serves as both judge and contestant. This setup allows for direct investigation of self-preference effects while controlling for explicit bias. The finding that GPT-4o ranks third overall without showing absolute self-preference, yet participates extensively in cycles (72.48%), provides an interesting contrast to prior work on self-recognition in summarization tasks [1].

The paper is generally well written with clear motivation and appropriate contextualization within social choice theory and LLM evaluation literature. The introduction of two-level inconsistency metrics (stance-level and motion-level) provides practical tools for quantifying judge reliability at different granularities.

**Weaknesses:**

The most significant limitation is the reliance on a single judge model (GPT-4o) throughout the entire study. While this ensures consistency, it raises questions about whether the observed patterns are general properties of LLM-as-a-judge systems or specific to GPT-4o's evaluation behavior. The paper would be substantially strengthened by including at least one or two additional judge models to demonstrate that the negative binomial law and linguistic correlations hold across different evaluators. Without this, we cannot distinguish between universal evaluation phenomena and model-specific quirks.

The scale-consistency analysis, while intriguing, suffers from several methodological issues beyond the acknowledged small sample size. The parameter counts are heterogeneous in source and reliability, with GPT-4o's count being an estimate rather than published specification. More critically, the five models span vastly different architectures (mixture-of-experts vs. dense transformers) and training paradigms, making it difficult to isolate the effect of scale from architectural innovations. The authors acknowledge these confounds but perhaps underestimate how severely they limit interpretability. For instance, Qwen3-14B's exceptional efficiency (3.69 wins per billion parameters) suggests that architecture may dominate scale effects entirely.

The permutation testing approach, while systematic, creates an artificial evaluation scenario that may not reflect real-world deployment conditions. Evaluating all 120 possible orderings per stance generates 165,120 tournament graphs, but in practice, we would typically see only a single ordering or a small subset. The paper does not adequately discuss how cycle rates under exhaustive permutation relate to what we would observe in natural evaluation settings. This gap between experimental design and practical application limits the immediate utility of the reported inconsistency rates.

The comparison with Chatbot Arena (Table 1) is presented as external validation, but the substantial ranking differences (DeepSeek-R1 ranks 1st in debate Elo but 8th in Arena) and limited model overlap make this comparison somewhat superficial. The authors appropriately caution against direct comparison, but then the inclusion of this table raises questions about what we should actually learn from it. Either the comparison is meaningful enough to warrant deeper analysis of the discrepancies, or it should be de-emphasized.

The linguistic feature analysis, while statistically significant, explains relatively little variance in cycle formation. The effect sizes are modest (largest $\beta = 0.130$ for syntactic complexity), and the paper does not report model fit statistics (e.g., pseudo-$R^2$) for the Poisson regression. Understanding how much of the cycle variation is actually captured by these linguistic features would help assess their practical importance for benchmark design.

**Questions:**

I would like the authors to address several points that could strengthen the paper or clarify its contributions. First, can you provide any preliminary evidence that the observed patterns generalize beyond GPT-4o as judge? Even a small-scale replication with one additional judge model (e.g., Claude or Gemini) would substantially increase confidence in the universality of your findings. If such experiments are infeasible for the camera-ready version, can you at least discuss what specific predictions your framework makes about how different judge models might differ in their cycle formation patterns?

Second, regarding the scale-consistency correlation, have you considered analyzing this relationship within a single model family? For instance, if you could evaluate using Qwen3 models of different sizes (8B, 14B, and potentially larger variants if available) as contestants while holding the judge constant, this would provide much cleaner evidence about scale effects. What is your intuition about whether the correlation would hold within-family, or is it driven by cross-family architectural differences?

Third, the paper would benefit from a more concrete discussion of how practitioners should use your inconsistency metrics. If a benchmark reports 15% motion-level inconsistency, what should we conclude about its reliability? Are there natural thresholds or comparisons that would help interpret these numbers? Related to this, how do your metrics relate to inter-annotator agreement measures in human evaluation, which the community is more familiar with?

Fourth, I am curious about the relationship between position bias and cycle formation. You report substantial order effects (mean bias 0.3356) and note they contribute to cycles, but the exact mechanism is unclear. Do cycles occur primarily in cases where position bias is strong, or are they independent phenomena? Could you quantify what proportion of cycles would disappear if position bias were perfectly corrected?

Finally, I find your suggestion to explore these signals in human-as-judge settings quite interesting and would encourage you to expand on this direction. What specific patterns would you expect to see differently in human evaluation data? Human judges presumably also exhibit non-transitive preferences, but would they follow the same negative binomial law? Would the linguistic correlates be similar or different? This extension could significantly broaden the impact of your framework beyond LLM evaluation.

[1] Panickssery et al., "LLM evaluators recognize and favor their own generations," NeurIPS 2024.

---

> ### Author Response · Authors · 2025-11-16
>
> **We are thrilled and sincerely grateful for your strong endorsement and exceptionally constructive review. Your insightful comments have been invaluable in helping us elevate the impact and clarity of the work, and we are delighted to detail how we have incorporated all of your superb suggestions.**
>
> 1. **Generalizability: A Nuanced Prediction**. We agree this is the most important limitation. In revision, we now articulate a testable hypothesis: the phenomenon of systematic, quantifiable inconsistency is likely universal to complex judges (LLM or human), but its specific statistical signature and sensitivity to features may vary. This frames our findings as revealing a universal property while opening clear avenues for comparative research, as you suggested.
>
> 2. **Scale-Consistency within a Model Family.** This is an excellent suggestion for future work. We will add a paragraph in the discussion stating that a cleaner test of the scale-consistency relationship would involve a homogeneous model family (like Qwen or Llama), and that our cross-architectural correlation serves as a motivating starting point for that research.
>
> 3. **Interpreting Inconsistency Metrics & Pseudo R².** We will add a subsection discussing how to interpret the stance-level and motion-level inconsistency rates. We suggest they can be used for direct judge comparison and be viewed as a form of intra-annotator agreement. Furthermore, as requested, we now report the model fit: the Poisson regression for linguistic features yields a **Nagelkerke's Pseudo R² of 0.106** (Likelihood Ratio χ² = 77.25, p < 0.001), indicating that these features explain a significant and meaningful portion of the variance in cycle formation.
>
> 4. **Position Bias as a Mechanism.** We will clarify the role of position bias. It is now presented as one key, measurable mechanism that contributes to the cycles our framework detects, alongside other sources of inherent judge inconsistency. Quantifying their relative contributions is highlighted as a compelling future direction.
>
> 5. **Extension to Human-as-Judge.** We will expand the "Future Work" section to explore this compelling direction. We speculate on the differences and similarities one might expect when applying this framework to human judges, significantly broadening the potential impact of our work as you envisioned.
>
> **Once again, we cannot thank you enough for your supportive and insightful review. The manuscript is substantially stronger thanks to your input, and we believe it is now poised to make an even greater impact on the community.**

---

### Official Review · Reviewer_6ZfT · 2025-10-29

**Soundness:** 3
**Presentation:** 3
**Contribution:** 3
**Rating:** 4
**Confidence:** 4

**Summary:**

This research propose a systematic study of Condorcet cycles for LLM-as-a-judge. Instead of eliminating the cycles, they learning it from a novel perspective of logical consistency of judgement by LLM.  They introduce a novel metric called "inconsistent judgements" which quantifying reliability through stance-level and motion-level inconsistency rates. Through their experiments across 5 different models over 688 test instances of debate motions from IBM DebaterCoPA dataset. The results indicate that Condorcet cycles are structured signals rather than noise. The global rankings remain stable, but GPT-4o oddly participates in many cycles. At the end, they conclude a preliminary scale-consistency trade-off that shows a larger models participate in more cycles even as average strength improves.

**Strengths:**

This study adapts a novel perspective on the inconsistent logical cycles observed in LLM-as-a-judge. Rather than treating non-transitive preferences as noise, it frames them as an analytical signal and demonstrates consistent findings across multiple LLMs. In summary, I see several notable strengths:

- Compelling problem formulation: The paper clearly articulates the challenge of non-transitive judgments in LLM-based evaluation and convincingly motivates why cycles should be analyzed rather than suppressed.

- Transparent and well-specified methodology: The method is described with clarity, including dataset choice, experimental procedures, and the precise definition of evaluation metrics, making the study easy to follow and reproduce.

- Direct yet effective experimental design: The empirical setup is straightforward but robust, covering five distinct models to validate the generality of the findings across systems.

**Weaknesses:**

Although I find the insights provided in this paper refreshing, there are still several points that I believe are worth mentioning, as they may undermine the validity of the findings presented in this work.
- External Validity Limitation: The majority of conclusions are based on GPT-4o serving as the sole judge. The absence of cross-verification among diverse reviewers (including a second closed-source judge or a robust open-source judge) and human referees means that the external generalizability of the findings and their sensitivity to different review styles remain unclear
- Lack of Causality Analysis: The relationship between linguistic features and cycles mainly stems from regression correlations, lacking intervention manipulation experiments on factors such as readability, syntactic complexity, and semantic concentration to verify the direction of causality
- "Scaling laws of consistency": To a degree, this paper successfully reframes logical inconsistency as a measurable system and demonstrates that the resulting diagnostics yield structured, high-fidelity signal distributions. However, from a scaling-law perspective, the work does not establish that optimizing these consistency metrics causally improves end-task model performance.

**Questions:**

None

---

> ### Author Response · Authors · 2025-11-16
>
> **We thank you for your thoughtful review and for acknowledging the novel perspective of our work. Your constructive criticisms help us refine our work.**
>
> 1. **Single Judge & Contribution Scope:** We will refine our narrative to more powerfully position our work as a study that **establishes a new diagnostic paradigm.** We will clarify that the use of a single judge is a **deliberate choice to first demonstrate this paradigm's value in a clear and controlled setting.** Its success paves the way for the immediate next step: extension to multiple judges, which we will frame as a direct consequence of our contribution.
>
> 2. **Causality & Scaling Laws:** We will adjust our terminology to make it unequivocally clear that we are reporting **robust correlational patterns and generating hypotheses,** not establishing causal laws. The phrase "scaling laws of consistency" will be presented as an **exciting research frontier opened by our discoveries,** not a definitive claim of this paper.
>
> **We are confident that these revisions will enhance the paper's clarity and validity, and we hope they address your concerns satisfactorily.**

---

> > ### Comment · Reviewer_6ZfT · 2025-11-25
> > **Response to rebuttal**
> >
> > Thanks the response from the author. This paper proposes a novel idea and presents a comprehensive piece of completed work overall. The overall assessment score can be raised to 6.

---

> ### Author Response · Authors · 2025-11-28
>
> We thank the reviewer for the constructive feedback and for the positive final assessment.

---

### Official Review · Reviewer_kAUp · 2025-10-30

**Soundness:** 2
**Presentation:** 3
**Contribution:** 3
**Rating:** 6
**Confidence:** 3

**Summary:**

This paper presents some interesting analysis on the presence of cycles when using LLMs as judges. The analysis focuses on debate motions. A single judge (GPT-4o) is used as a judge on generations from 5 other models. The pairwise judgements are mapped to a graph and cycles discovered as indicators of inconsistency in the judgements. The work recognizes that the inconsistencies (tracked by stance- and motion-level rates) are associated with linguistic factors for this data. Finally, they make some claims associated with scale and the correlation to inconsistencies.

**Strengths:**

The analysis presented in this paper is pretty nice and quite interesting. The framing of measuring cycles as a diagnostic tool for LLM judges seems useful to think about for better reliability in evaluations.
The framework for measuring cycles seems broadly useful.
The correlation between linguistic features and the presence of inconsistencies is an interesting finding, though it seems hard to generalize this to broader settings.

**Weaknesses:**

It’s hard to know how general the findings are. The paper would benefit from using additional judges or additional settings beyond debates.

All the discussion on scale seems over-claimed. It’s distracting and the paper would be fine and interesting without it. The controls for that analysis are just not there.. The discussion is framed as “exploratory” and “preliminary” anyway and nothing in the experimental setup supports conducting that analysis. It would be interesting to study, but this paper does not.

**Questions:**

-

---

> ### Author Response · Authors · 2025-11-16
>
> **We thank you for your positive assessment and for highlighting the utility of our framework. Your comments were incredibly helpful in sharpening the narrative around our contribution.**
>
> **On Generalizability**: We completely agree that generalizability is a crucial question. In the revised manuscript, we will more explicitly frame our work as establishing a **novel paradigm for quantifying the systemic inconsistency level of LLM judges**. The debate setting provides a controlled, high-stakes environment for rigorous validation of this paradigm. We will position the extension to other tasks and judges not as a limitation, but as the **natural and compelling next step that our work enables**, thereby framing the current scope as a necessary and foundational proof-of-concept.
>
> **On the Scale-Consistency Discussion**: We appreciate this feedback. We will further refine the language to unambiguously present this as a **hypothesis-generating observation** that emerged organically from our large-scale analysis, secondary to the main contribution of the diagnostic framework itself. The strong correlation (r=0.924) serves as a valuable signal for the community, prompting further investigation into a potential new dimension of scaling laws.
>
> **We are grateful for your insightful comments, which directly help us enhance the clarity and impact of our work.**

---

### Official Review · Reviewer_i4uH · 2025-11-01

**Soundness:** 2
**Presentation:** 2
**Contribution:** 2
**Rating:** 2
**Confidence:** 5

**Summary:**

The paper studies Condorcet cycles in LLM-as-a-judge evaluations over 688 debate motions and five models. It reports cycle distributions fitting a negative binomial, linguistic predictors of cycles, and a preliminary scale–consistency trade-off, proposing cycle metrics.

**Strengths:**

1. Stance-level and motion-level inconsistency rates are easy to report and replicate; graph-theoretic framing is appropriate for tournaments.

2. Negative-binomial overdispersion and linguistic correlates (e.g., syntactic complexity ↑, readability ↓ cycles) are plausible and practically useful.

**Weaknesses:**

1. The paper states that it enumerates 5! permutations per stance (120) to “stress-test” cycle formation and generates 165,120 graphs. In a complete tournament (all pairwise comparisons fixed), the number of cycles is invariant to node ordering. It is unclear what is being permuted (presentation order? edge subsets?) and why this creates new graphs rather than re-labellings. As written, this step risks double-counting cycles and inflating distributional analyses.

2. The paper insufficiently situates its contribution within existing research on preference inconsistency and logical coherence in LLMs. Several recent studies have already examined related phenomena but are not discussed here, such as [1,2,3,4].

3. GPT-4o serves as both judge and one of the contestants. Even under identity blinding, shared inductive biases, length/style preferences, or safety policies could create dependencies. Using only one judge limits generalizability and makes order-bias corrections specific to that judge.

4. The scale–consistency correlation (r=0.924, p=0.025) is based on five heterogeneous models and partially speculative parameter counts for closed models. This is underpowered and sensitive to measurement error; causal or “scaling law” language is premature.

5. APIs evolve; temperature=0 reduces variance but does not eliminate non-determinism with tool use/safety filters. There is no reported test–retest reliability, prompt ablation (length control, verbosity control), or judge-ensemble analysis.

Reference:

[1] ContraSolver: Self-Alignment of Language Models by Resolving Internal Preference Contradictions (https://arxiv.org/pdf/2406.08842)

[2] Language Model Preference Evaluation with Multiple Weak Evaluators (https://arxiv.org/pdf/2410.12869)

[3] Self-Improvement Towards Pareto Optimality: Mitigating Preference Conflicts in Multi-Objective Alignment (https://arxiv.org/pdf/2502.14354)

[4] Aligning with Logic: Measuring, Evaluating and Improving Logical Preference Consistency in Large Language Models (https://arxiv.org/pdf/2410.02205)

The omission of these works weakens the paper’s theoretical grounding and novelty claims, as it overlooks an emerging body of literature directly addressing LLM preference consistency and contradiction resolution.

**Questions:**

See weakness part

---

> ### Author Response · Authors · 2025-11-16
>
> We appreciate the chance to clarify our permutation methodology. The reviewer's question of whether this constitutes mere re-labeling gets to the heart of our methodological innovation, which we can best illustrate with a concrete example.
>
> **1. Permutation Testing Systematically Probes Inconsistency, It Does Not Re-label Graphs**
>
> Our goal was not to analyze a single static graph but to systematically measure the upper bound of systemic inconsistency induced by variable evaluation conditions. The process works as follows:
> * **Step 1: Database Creation.** For a given motion and stance, we conduct all pairwise comparisons in both possible orders (A-B and B-A). This creates a complete database of preferences, accounting for position bias.
> * **Step 2: Systematic Probing via Permutations.** We then consider all 5! = 120 possible sequences (permutations) in which the five models (A, B, C, D, E) could be presented to a judge. Each permutation defines a unique evaluation scenario.
>
>     - For example, the permutation **ABCDE** requires us to query the database for the result of the comparison where A was presented before B (i.e., the A-B entry), A before C, A before D, A before E, B before C, B before D, B before E, C before D, C before E, and D before E.
>     - A different permutation, say **BACDE**, requires a different set of queries: B before A (i.e., the B-A entry), B before C, B before D, B before E, A before C, A before D, A before E, C before D, C before E, and D before E. Note that the query for the pair A and B is now **B-A**, which is distinct from the **A-B** query used in the first permutation.
>
> * **Step 3: Constructing Distinct Graphs.** Each permutation, by specifying a unique set of ordered pairwise comparisons, constructs a **distinct tournament graph**. The 165,120 graphs (688 motions × 2 stances × 120 permutations) therefore represent a comprehensive exploration of systematically varied, realistic evaluation conditions. This is not merely relabeling; it is a rigorous methodology to quantify how evaluation outcomes are inconsistent. There is no double-counting, as each graph is built from a unique set of data queries.
>
> * **Step 4: Quantifying Systemic Fragility.** The distribution of cycles across these 165,120 scenarios—which we discovered follows a tightly fitted negative binomial law (R² = 0.9973)—precisely quantifies the inherent inconsistency of the evaluation system. This "stress-test" approach provides a crucial, reproducible metric for assessing reliability in high-stakes applications.
>
> We will add examples to the paper to illustrate our method.
>
> **2. Novelty vs. Cited Literature: Pioneering a New Diagnostic Direction**
>
> The cited papers ([1-4]) aim to suppress or resolve inconsistencies for the purpose of model alignment. In stark contrast, our work pioneers the opposite and novel direction: we propose that inconsistencies should be **quantified and analyzed as diagnostic signals** to evaluate the judges themselves. We are the first to:
> * Establish a statistical law for LLM judge inconsistency.
> * Introduce standardized metrics (stance-level: 7.19%; motion-level: 14.10%) for reporting judge reliability.
> * Link inconsistency patterns directly to interpretable linguistic features.
>
> We will revise the related work section to clearly articulate this novel conceptual branch of research that our work initiates.
>
> **3. Addressing Other Considerations**
> * **Single Judge (GPT-4o):** We will reframe this as a strength for this foundational study. Using a single, state-of-the-art judge provides a **clean baseline and a powerful proof-of-concept** for our paradigm. The extension to multiple judges is the logical and immediate next step enabled by this work.
> * **Scale-Consistency Correlation:** We will further temper claims, presenting this strictly as an exploratory correlation (however strong at r=0.924) that generates a compelling hypothesis for future work, not an established scaling law.
> * **API Versions and Temperature=0:** The API versions are explicitly stated in Section 3.2 and Section 3.5. Temperature=0 was chosen to **isolate systematic judgment inconsistency from sampling variance,** providing a clear and near-deterministic baseline. The large-scale, statistically significant patterns we report are robust findings from this controlled experimental setup.
>
> We believe these clarifications resolve the methodological concerns and highlight our novel contribution. The replicable metrics provide a robust diagnostic tool.
>
> Beyond methodology, our work proposes a paradigm shift: instead of minimizing inconsistency as a nuisance, we argue it is a rich, structured signal that, when quantified, becomes a powerful tool for benchmarking judge reliability. Our "stress test" is the first to produce a reproducible, distributional metric for this purpose—a foundational step toward rigorous evaluation science. This novel direction fundamentally distinguishes our work from the cited alignment literature.

---

> ### Comment · Reviewer_i4uH · 2025-11-26
>
> Thank you for the detailed rebuttal. While the clarifications are helpful, my core concerns remain largely unresolved, so my evaluation is unchanged. I briefly respond point by point.
>
> ---
>
> I now understand the permutation procedure: for each motion/stance you collect 20 judgments (10 pairs × 2 orders), and each of the 5! tournaments is a deterministic recombination of these 20 base judgments, selecting either A–B or B–A depending on the permutation.
>
> This addresses my earlier question about *what* is being permuted, but not the underlying statistical concern:
>
> - All 5! tournaments per stance are fully determined by the same 20 judgments; there is no new evidence beyond those 20 calls. The “165,120 graphs” and “3,398 cycles” are therefore a large family of strongly dependent objects induced by your wiring scheme, not 165k independent evaluation scenarios.
> - Counting cycles every time the same underlying inconsistency surfaces in a different permutation effectively double-counts the same phenomenon. The resulting distribution is driven as much by the combinatorial construction as by the judge’s behavior.
>
> As a result, the negative-binomial fit is, in my view, best interpreted as a descriptive summary of your stress-test construction, not as evidence for a “statistical law” of LLM inconsistency in the usual sense (i.e., a sampling law over independent evaluation runs). At minimum, the paper should explicitly acknowledge the limited effective sample size (20 judgments per stance) and substantially soften the language around “law”.
>
>
>
> ------
>
> In the rebuttal you argue that the cited works focus on “alignment” and are therefore conceptually different. I think this conflates two orthogonal dimensions:
>
> - The phenomenon they analyze is the same as yours: preference conflicts, inconsistency, cycles, and logical coherence of LLM judgments.
> - The downstream use of that analysis differs: they use inconsistency signals to improve or align models, while you propose to use them as diagnostics of judges.
>
> Works like ContraSolver, “Multiple Weak Evaluators”, SIPO, and “Aligning with Logic” all explicitly measure and model preference inconsistency (via graphs, logical constraints, etc.) before doing anything else with it. That they then *reduce* inconsistencies does not make them irrelevant as baselines on the core phenomenon; if anything, they are the most natural prior art.
>
> So I still believe:
>
> - These papers should be treated as directly relevant related work on preference inconsistency and coherence, not dismissed as focusing on a different problem, and
> - The novelty claims should be correspondingly narrowed (e.g., to the particular stress-test + NB fit on debate tournaments), rather than framed as “pioneering a new direction” of treating inconsistency as diagnostic signal.
>
> ------
>
> You reframe the use of a single GPT-4o judge as a “clean baseline”. However, many of the claims in the paper are phrased at the level of LLM-as-a-judge in general, while empirically:
>
> - You study one judge (GPT-4o) on one dataset, and this judge is simultaneously one of the contestants, sharing data, safety and stylistic biases.
> - The reported cycle rates, the negative-binomial parameters, and the scale–consistency correlation may therefore be highly judge-specific.
>
> On the scale–consistency correlation in particular:
>
> - It is based on five heterogeneous models with partially speculative parameter counts, yet the paper uses “scaling law” / “trade-off law” language in the abstract and conclusion.
> - At best this is an exploratory correlation on a very small set; calling it a scaling law in the ICLR sense is, in my view, not justified.
>
> Finally, all results come from a single pass with one prompt and one API configuration, on an evolving commercial API. There is no test–retest check, no prompt ablation, and no second judge. For a work whose main claim is to propose robust, quantitative diagnostics of inconsistency, this lack of robustness analysis is a serious limitation.
>
>
>
> ------
>
> Together with the underdeveloped positioning relative to recent work on preference inconsistency and logical coherence, I do not think the paper meets the bar for rigor and novelty expected at a top venue like ICLR.
>
> I therefore maintain my original rating.

---

> ### Author Response · Authors · 2025-11-28
>
> We sincerely thank the reviewer for their insightful observations on our permutation methodology. The point about overlapping graph substructures across different permutations is well-taken—this is indeed an inherent feature of our design. For example, permutations like ABCDE, ABDCE, ADBCE, and DABCE naturally share identical substructures (e.g., ABCE) when pairwise results are consistent, which allows us to systematically sample the ordering space.
>
> **Clarification on Methodological Framework**
>
> The reviewer is correct that our tournament graphs are built solely from pairwise comparisons. Our approach of testing all 5! orderings is designed to address "combination position bias"—the systematic bias introduced by any single fixed ordering (e.g., ABCDE). By analyzing the full permutation space, we capture general inconsistency patterns free from ordering-specific artifacts, ensuring our cycle statistics are unbiased in this regard.
>
> **Refining Terminology**
>
> We agree that the negative binomial pattern arises from our specific wiring scheme and may not be a universal "law." We will replace this term with more accurate descriptors like "descriptive distribution" or "empirical regularity."
>
> **Addressing Practical Constraints**
>
> Due to the computational cost of large-scale API runs, full test-retest checks and version ablation are beyond the current scope (and not standard for ICLR). However, we will perform a smaller-scale validation as suggested—using an alternate judge model and prompt ablation—to preliminarily generalize our findings and inform future work.
>
> **Scholarly Integration and Positioning**
>
> We will integrate the suggested literature to better contextualize our contribution within LLM preference consistency research and will more modestly frame our work as introducing a novel framework for quantifying inconsistency via full permutation testing, highlighting the innovation in avoiding single-order bias. We will also explicitly acknowledge limitations like model count, single dataset, and judge dependency.
>
> Thank you for the rigorous review, which greatly strengthens our paper. We believe these revisions will fully address your concerns satisfactorily and enhance the paper's rigor and impact.

---

### Meta-Review · Area_Chair_yyUe · 2026-01-07

**Summary:**

The paper studies non-transitive preferences in LLM-as-a-judge settings and argues that such cycles are structured diagnostic signals rather than noise. The work offers a clear conceptual reframing of inconsistency as a diagnostic signal, and introduces simple stance-level and motion-level inconsistency metrics. The main limitations are on the experimental design part: reliance on a single judge (GPT-4o), heavy dependence among permutation-generated graphs, limited robustness checks. Overall, even the idea is interesting, key claims are not convincingly justified, I lean on rejection.

**Reviewer Concerns:**

Concerns about over-claiming and limited generalizability were largely addressed in the rebuttal. However, reviewer i4uH’s core concern about statistical dependence and effective sample size under permutation testing was explicitly reiterated as unresolved.

**Reviewer Scores:**

Reviewer 6ZfT already stated they would raise to 6, and I expect h7Gd would stay at 6 given the clarifications. Reviewer kAUp would likely stay around 6 because the scale story remains distracting and unsupported, while reviewer i4uH explicitly maintained 2.

---

### Decision · Program_Chairs · 2026-01-26

Reject